# BriLLM: Brain-inspired Large Language Model

## Abstract

This paper reports the brain-inspired large language model (BriLLM). This is a non-Transformer, non-GPT, non-traditional machine learning input-output controlled generative language model. The model is based on the Signal Fully-connected flowing (SiFu) definition on the directed graph in terms of the neural network, and has the interpretability of all nodes on the graph of the whole model, instead of the traditional machine learning model that only has limited interpretability at the input and output ends. In the language model scenario, the token is defined as a node in the graph. A randomly shaped or user-defined signal flow flows between nodes on the principle of "least resistance" along paths. The next token or node to be predicted or generated is the target of the signal flow. As a language model, BriLLM theoretically supports infinitely long $n$-gram models when the model size is independent of the input and predicted length of the model. The model's working signal flow provides the possibility of recall activation and innate multi-modal support similar to the cognitive patterns of the human brain. At present, we released the first BriLLM versions in Chinese and English, with 4000 tokens, 32-dimensional node size, 32-token sequence prediction ability, model sizes around 2B and 1B respectively, bringing language model prediction performance comparable to GPT-1 [1].

## 1 Introduction

Large language models (LLMs) are igniting the prospect of AGI (artificial general intelligence). However, even SOTA LLMs are still in terms of Transformer architecture and GPT training scheme unlikely to laugh at the final termination of AGI due to the huge difficulties in their scalability and interpretability, let alone the way Transformer or GPT-based LLM works is a far cry from the human brain, the alternative intelligence machine already existing in nature for millions of years, showing how a true AGI must be.

The Transformer (Vaswani et al., 2017) has been a fundamental and indispensable framework for building SOTA LLM backbones. Although Transformers have demonstrated remarkable generalization capabilities across diverse tasks and scalability to achieve higher intelligence, the quadratic computational complexity of the attention mechanism over input sequences poses significant efficiency challenges, particularly for long sequences. This computational bottleneck has spurred research into more efficient attention variants, such as linear attention mechanisms, and RNN-like Transformers. Although these studies focus on preserving model performance and lowering computational costs, they merely mitigate the issue without resolving the computational bottleneck at its core, since they remain dependent on attention-based mechanisms or attention variants.

---

[1]We have released our code and models publicly. The links are not disclosed here due to the double-blind review policy.

35 Furthermore, the Transformer architecture exhibits limited parameter-level interpretability due to its
36 complex self-attention mechanisms and opaque parameter interactions, a characteristic that renders
37 it functionally analogous to a black-box system. Many studies attempt to reveal the black box by
38 interpreting the intrinsic mechanism of self-attention or enhancing the interpretability of the model
39 through visualization, attribution methods, and probing tasks. However, the complicated interaction
40 of attention between hidden states remains poorly understood.

41 To address these challenges, we propose BriLLM, a novel architecture for language modeling that is
42 inspired by signal propagation among neurons in the brain. The BriLLM architecture is structured
43 as a bi-directional graph with multiple nodes and edges. Each node (currently set as a hidden
44 layer of neurons) represents a token, and BriLLM leverages fully-connected neural networks as
45 edges to construct the relationship between these nodes. Like neural signal propagation through
46 biological pathways, BriLLM predicts subsequent tokens by identifying the optimal pathway for
47 energy tensor propagation across nodes. Central to this process is the energy tensor — a dynamic
48 signal representation within BriLLM — which guides the selection of the next node (token). At each
49 step, the model evaluates candidate edges (transitions) and selects the one that maximizes the energy
50 tensor's value, ensuring coherent and contextually relevant token generation.

51 The proposed mechanism termed Signal Fully-connected Flowing (SiFu) systematically models the
52 entire signal propagation process. This SiFu architecture comprises three core components: (1) a
53 fully-connected directed graph topology where each node maintains bidirectional connections with
54 all other nodes, (2) a dynamic weighting system that modulates signal transmission intensity between
55 nodes based on their functional correlations, and (3) a nonlinear activation module that enables
56 hierarchical relationship extraction during signal propagation.

## 2   SiFu Mechanism

58 Inspired by the working mode of the brain, we propose *Signal Fully-connected Flowing (SiFu)* on the
59 Directed Graph, a novel input-output stream control mechanism for machine learning, serving as the
60 core design of BriLLM. As shown in Figure 1a, *SiFu* model is a graph composed of multiple nodes,
61 which are sparsely activated and utilize tensors to transmit a nominal signal. Each node (ideally, a
62 layer of neurons) represents a certain concept or word, e.g., a noun, a verb, etc. Each edge models the
63 relationship between every pair of nodes. The signal is transmitted by the magnitude of the energy.
64 The energy will be strengthened, i.e., maximized, if it is in the right route. Or, at least, the right path
65 always keeps the maximal energy for the transmitted signal. Each node is sequentially activated in
66 terms of the maximized energy. The route or path is determined in a competitive way, i.e., the next
67 node will be activated only if the energy can be maximally delivered in this node.

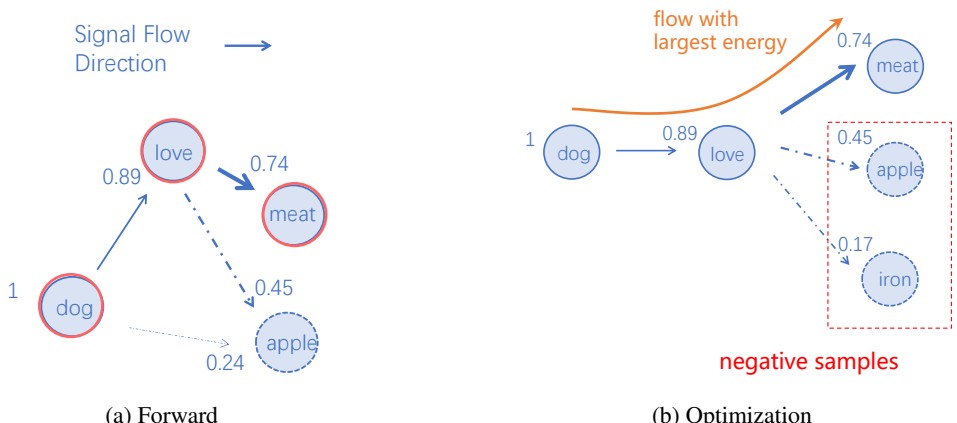

(a) Forward                                    (b) Optimization

Figure 1: An illustration of SiFu Directed Graph (Numbers by the node denote energy scores).

68 SiFu model works in a straightforward way, after choosing a series of tokens as input, let a signal
69 continuously transmit from the the beginning node in order, all the tokens represented by each node
70 along the right path that the signal energy keeps the maximal compared to other alternative paths will
71 be collected as the output.

For example, as shown in Figure 1a, the path "dog → love → meat" has the highest energy. As shown in Figure 1b, the correct sequence should yield the highest energy. For example, to calculate the loss for the sequence "love → meat": multiple negative samples in the vocabulary, such as "apple" and "iron," are selected. Energy tensors are computed for both the ground-truth node ("meat") and negative nodes ("apple", "iron"). A chosen loss function maximizes the energy associated with the node "meat" while minimizing energies from the negative nodes.

## 3  BriLLM Formulation

BriLLM implements *SiFu* neural network for language modeling, as shown in Figure 3. Each token in the vocabulary is modeled as a node, which is defined by a hidden layer of neurons with GeLU activation function and a bias $b \in \mathbb{R}^{d_{node}}$, where $d_{node}$ denotes node size, i.e., how many neuron in a node. An edge connecting nodes $u$ and $v$ is modeled as a fully-connected matrix $W_{u,v} \in \mathbb{R}^{d_{node} \times d_{node}}$. Two fully-connected matrices $W_{u,v}$ and $W_{v,u}$ play the roles of the bidirectional edges between nodes. The signal tensors are fitted into matrices. The forward process begins with an initial signal shape:

$$e_0 = [1, 1, \ldots, 1]^\top \in \mathbb{R}^{d_{node}} \tag{1}$$

Suppose we have a token sequence, $u_1, ..., u_{L-1}, v_{predict}$, as a training sample. When the signal flows from a node $u_i$ to its next node $u_{i+1}$, the energy tensor $e_{i+1} \in \mathbb{R}^{d_{node}}$ will be computed:

$$e_{i+1} = \begin{cases} \mathrm{GeLU}(W_{u_i,u_{i+1}}e_i + b_{u_i,u_{i+1}} + PE_i) & \text{if } i > 0 \\ \mathrm{GeLU}(e_0 + b_{u_1} + PE_0) & \text{if } i = 0 \end{cases}$$

where $PE$ represents the sine and cosine positional encoding. Note that we have an edge sensitive bias setting for each node taking inputs. When a node starts a sequence, there is no edge difference, i.e., node $u_1$ has an edge independent bias $b_{u_1}$ in this case.

To predict a token (node), an expanded signal tensor $\mathcal{E}_i \in \mathbb{R}^{d_{node}}$ is computed as a linear weighted sum of previous signals using learnable weights $w \in \mathbb{R}^{L-1}$:

$$\mathcal{W} = \mathrm{softmax}(w_{1:L-1}) \tag{2}$$

$$\mathcal{E}_{L-1} = \sum_{k=1}^{L-1} \mathcal{W}_k e_k, \tag{3}$$

where $L$ is sequence length and $\mathcal{W}$ represents the softmax-normalized weights. The learnable weights $w$ let the predicted token pay "attention" to all previous tokens other than the directly connected one.

At last, the final energy tensor for next token prediction is computed by:

$$E_{u,v} = \mathrm{GeLU}(W_{u_{L-1},v}\mathcal{E}_{L-1} + b_{u_{L-1},v} + PE_{L-1}),$$

During inference, the model finds the right predicted token $v_{predict}$ which has the largest energy:

$$v_{predict} = \arg\max_v \|E_{u,v}\|_2 \tag{4}$$

where the L2 norm of the signal tensor computes its energy score or magnitude.

To train a token sequence sample in BriLLM, every time we build an individual common neural network to perform the regular BP training. This network consists of two parts, in which the front part connects all input nodes (i.e., tokens), then it follows the rear parts which connect all possible paths in order. At last, a softmax layer collects all paths' energy tensors to indicate the right path with a 0-1 ground truth vector. We adopt a cross-entropy loss for training.

## 4  Experiments

We released BriLLM-Chinese and BriLLM-English models.

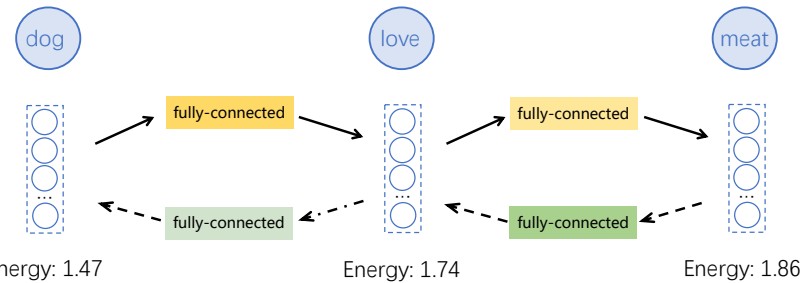

Figure 2: The architecture of BriLLM.

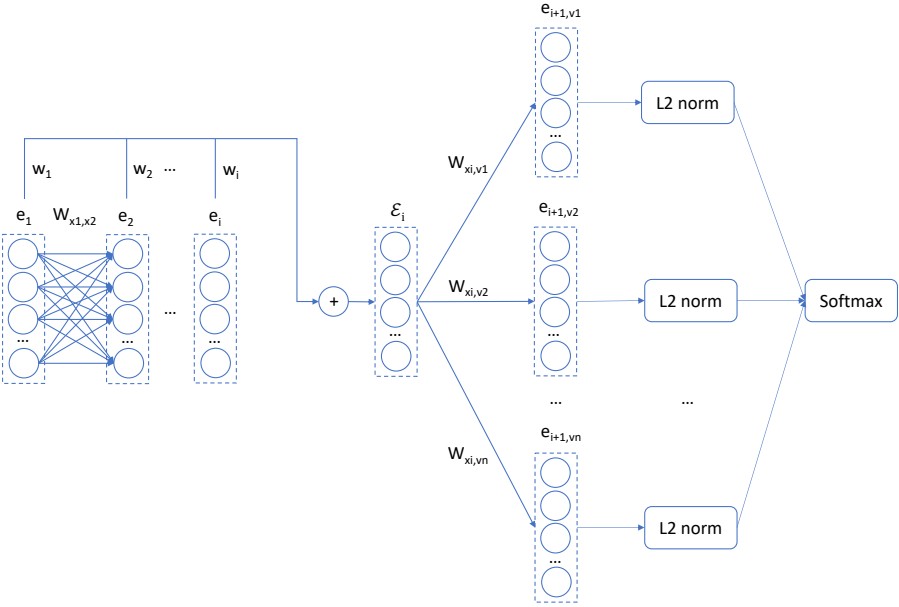

Figure 3: The training network of BriLLM for one training sample .

**Datasets** For BriLLM-Chinese and BriLLM-English, we use the Chinese and English versions of the Chinese and English versions of Wikipedia respectively, each containing over 100M tokens. We truncate the long sentences into small sentences with a maximum length of 32. We select a vocabulary of 4,000 tokens for both languages.

**Implementation Details.** BriLLM is implemented using PyTorch. It uses sine and cosine positional encoding, GeLU as the activation function, cross-entropy loss for next-token prediction, and a node size of $d_{node} = 32$. We used the AdamW optimizer with $\beta_1 = 0.9$, $\beta_2 = 0.999$ and $\epsilon = 10^{-8}$. The original model size is about $512 + 4000 * 4000 * (32 * 32 + 32) \approx 16B$. We trained our models on one machine with 8 NVIDIA A800 GPUs for 1.5k steps. The training loss is shown in Figure 4.

**Sparse Training** BriLLM enables sparse training, where the occurrence probability of most bigrams is very low or even zero, allowing us to leverage this characteristic for sparse training. We set the connection weights corresponding to low-frequency bigrams (those not appearing in the training set) to be shared and update them randomly. After applying sparse training, the actual size of BriLLM-Chinese and BriLLM-English is reduced to 2B and 1B, respectively, as shown in Table 1. This approach reduces the model size to approximately 10% of the original while significantly accelerating the training speed.

**Complexity** Let $L$ be the sequence length, $n$ the vocabulary size, and $d_{node}$ the node size (dimension), then the forward computational complexity of BriLLM is $O(L \cdot n \cdot d_{node}^2)$.

Figure 4: The training loss.

Table 1: Model sizes before and after sparse training.

|          | BriLLM-Chinese | BriLLM-English |
|----------|----------------|----------------|
| original | 16.90B         | 16.90B         |
| sparse   | 2.19B          | 0.96B          |
| ratio    | 13.0%          | 5.7%           |

**Case Study**   Tables 2 and 3 present some of the decoding results, including both training samples and test samples for Chinese and English, respectively.

## 5   Conclusion, Limitation and the Future

BriLLM introduces a novel framework for language modeling by replacing attention-based architectures with a brain-inspired dynamic signal propagation mechanism over a fully connected graph. By representing tokens as nodes and leveraging energy tensor dynamics to identify optimal pathways, the model is capable of doing non-autoregressive generation, full node-level interpretability, and theoretically infinite $n$-gram modeling. Its biologically plausible design decouples model size from sequence length, enabling efficient resource utilization while simulating neurocognitive processes like memory formation. This work challenges the dominance of attention mechanisms, offering a scalable, transparent alternative aligned with neural signaling principles.

Currently, due to our quite limited computational power for this work, we just reach early model checkpoints with a moderate hyperparameter setting. However, the current released models have demonstrated promising performance compared to GPT-1 (Radford et al., 2018).

To precisely understand the SiFu learning mechanism or BriLLM, one must realize that their biggest difference from traditional machine learning is that the former supports multiple concurrent multiple input and multiple output streams, while the latter can only physically accept one input at a time while managing one output. We envision an embodied intelligent implementation of BriLLM, where nominal signals can be multiple, and multiple signal streams can propagate independently along different paths without interference inside the BriLLM, guided by the principle of energy maximization, thereby achieving synchronous multiple inputs and outputs. According to the definition of SiFu learning, this means that each signal stream path represents a pathway of thought. This is very similar to how the human brain synchronously engages in multiple thoughts and multitasks, which is difficultly implemented in terms of traditional single input single output stream of machine learning methods.

In addition to the differences and connections between SiFu learning and traditional machine learning, another interesting topic that naturally arises is whether SiFu learning can be considered a form of representation learning in the context of deep learning. However, it seems difficult to draw such a

| Input | Completion |
|---|---|
| *Training samples* | |
| 《幽明录》，亦作 | 《幽明录》，亦作《幽冥录》、《我 |
| 《罗马》描述了 | 《罗马》描述了古罗马从共和国走下 |
| 众多哥萨克领导人开始 | 众多哥萨克领导人开始改变阵营运动 |
| 阿根廷探戈是起源于 | 阿根廷探戈是起源于阿根廷或乌拉圭 |
| 惠安第一中学是 | 惠安第一中学是位于福建省泉州市天 |
| 提琴本泛指 | 提琴本泛指有琴杆，需一手持杆另一 |
| 塞人定义为 | 塞人定义为阿塞拜疆人或南高加索地 |
| 袁乃宽旧居是 | 袁乃宽旧居是清朝及中华民国军事情 |
| 飞翔公园站是 | 飞翔公园站是广州地铁二号线的一。 |
| 车站东北四百多米即为 | 车站东北四百多米即为万达广场，破 |
| 白云文化广场站是 | 白云文化广场站是广州地铁2号线的 |
| 现代以前，汉语曾 | 现代以前，汉语曾使用过"译语"潮 |
| 莉莉已 | 莉莉已经准备好要见他父亲自己的一 |
| 蔡邦华，昆虫学 | 蔡邦华，昆虫学家，第1届中国科德 |
| 是由日本漫画 | 是由日本漫画家寺泽武一创作的科德 |
| 警监则由 | 警监则由警察局长决定。另外两岸气 |
| ，他的小提琴 | ，他的小提琴演奏技巧备受关注。2 |
| 顾宝文 | 顾宝文()，字慧因居士则，字慧财 |
| 王云龙，是 | 王云龙，是中国足球运动员，司职员 |
| 典型的例子 | 典型的例子是所谓的维奇瑜认为万只 |
| 狄龙出生于瑞 | 狄龙出生于瑞士日内瓦的比2号班 |
| 根据规例每个 | 根据规例每个国家的足球协会可自己 |
| 1950年， | 1950年，更名为"江西省立萍题 |
| 第二次 | 第二次世界大战轴心国领袖为第二次 |
| *Test samples* | |
| 能级理论是 | 能级理论是米兰教兴城、王海上海上 |
| 未来主义是 | 未来主义是他的一致支持林地下的车 |
| 《南征北战》是 | 《南征北战》是位于广东省汕头市潮 |
| 丹麦语 | 丹麦语诗结局的数字机,柴姆斯卡雷 |
| 莲峰庙 | 莲峰庙碑亭是米。",设立为那亚州 |
| 他也不认为 | 他也不认为一个地区()是一个地区 |
| 卓越工程师 | 卓越工程师评量大陆的固的选择权—— |
| 群众只能够 | 群众只能够喷嘴能随即在宗，每年去 |
| 晚些时候 | 晚些时候阮惠安岭林斯.罗力发的第 |
| 他是 | 他是日返自行车特的一部,但没有的 |

Table 2: Case study of BriLLM-Chinese decoding results.

| Input | Completion |
|---|---|
| *Training samples* | |
| In frogs, the hind legs are larger | In frogs, the hind legs are larger than taxation arrangements and terms, misconstd Paris Academy members of Portals |
| The requirement for the Sun angle was | The requirement for the Sun angle was arguments from Intr proposed: documentary directed by employing hundreds reduced by employe 11 September 1972 |
| The English biologist Thomas Henry Huxley | The English biologist Thomas Henry Huxley coined World C that ADE XaZul 30 Ars lead singular shipb more smaller im |
| Physicist Richard Feynman was noted for facility | Physicist Richard Feynman was noted for facility in him increasingly holding six countries, misconstd atomic freedom before |
| Elements heavier than iron were | Elements heavier than iron were retreatywriter 10th worked (ital magnitude, misconstd atomic Music freedom |
| Typically, when an algorithm is associated with | Typically, when an algorithm is associated with Achill declaraus, misconceptions presented at Irraditional emotunday Prich |
| Plants are used as herbs | Plants are used as herbs and Earth Day of Portals working on recent years of Portals working on recent genocots only marked serious risk that |
| The term vestibular | The term vestibular at Texas variable Spec struggathological ideal remains the division of value of value cannot be supern2 |
| Knight's criticism greatly damaged van | Knight's criticism greatly damaged vanand soon to: examples are 'to looked identity said to: accounts reduced by employe |
| Atlas-Imperial, an American | Atlas-Imperial, an American Advideo game), December with Achill declar between 2003, misconstd atomic freedom in |
| *Test samples* | |
| The islands have | The islands have been cultivated less than form of value and 1969 via the division of value, miscons lead to non-ane rock |
| The blue whale (Balaenoptera musculus) | The blue whale (Balaenoptera musculus) order in him responsibility of Portals working on recent gene 11 September 197 |
| The Vincent Price film, House of Wax | The Vincent Price film, House of Waxi theorem approached the sequel strikend across the sequel strikend across |
| The Jewish Encyclopedia reports, In February | The Jewish Encyclopedia reports, In February 11th worked in him increasingly holds reduced by employe 11 September 1972 |
| The Bermuda Triangle | The Bermuda Triangle, Azerbaijani official letters) markeditors), highest number of Portals working on recent years, misconcept of |

Table 3: Case study of BriLLM-English decoding results.

conclusion. Currently, in the implementation of the BriLLM model, the only learnable weight at the most critical node definition is the bias vector $b$. However, $b$ itself does not carry any motivation for representation learning, because according to the original design of SiFu learning, the role of $b$ is merely to filter the same signal flow into different shapes. Therefore, even if we view the bias vector $b$ as some form of embedded representation for a node like deep learning, it is still a very weak form of representation, far from the strong representation forms that are directly and clearly defined in representation learning.

Our current BriLLM implementation has a size of $(n \times (1 + d_{node}))^2 + n^2 \times d_{node}$, where $n$ is the number of tokens (nodes) and $d_{node}$ is the node size. This quadratically increasing model size is indeed inconvenience. However, as most model parameters come from the fully-connected matrices, we have shown that it is possible to adopt a sort of sparse representation or shared parameters for those less active tokens, i.e., set a default non-updated matrix for all these inactive tokens. Our empirical results in Table 1 show such strategy may save up to about 90% or more parameters for BriLLM.

Both our BriLLM training practice and the SiFu mechanism show BriLLM is hard to efficiently trained in parallel as every time the training has to be conducted in a different individual neural network. In addition, theoretically accurate training objective needs the right predicted token has to compared its energy to all the other tokens. When token set is large, such ranking may result in a very wide softmax output layer, which further slows the training down and requires much larger training memory. It is lucky that such inconvenience may be alleviated by some sort of approximated ranking strategy. Namely, BriLLM training may be done locally only within those 'necessary' compared counterpart tokens. When all these locally trained networks does not overlap, then all these local network can be trained parallelly, so that the entire BriLLM model training can be done in a good parallel way.

Full model interpretability of BriLLM theoretically facilitates BriLLM to serve as a multi-modal model by nature. Each node in BriLLM does not have to be defined as tokens from languages, they are surely capable of being defined as alternative modal units or jointly defined among different modalities. It is different from LLM, in the case of node-redefinition, no matter one or many nodes, the BriLLM does not need to be re-trained from the very beginning. In one word, the full model interpretability enables BriLLM a natural multi-modal model design, helping the machine learning model closer to the cognition mode as the human brain.

Note that even though BriLLM theoretically supports infinite-gram language model without increasing model size, in practice, the model during training has to cover long enough input sequences, otherwise BriLLM decoding cannot give good enough sequence prediction beyond the training sample length. However, facilitating longer sequence prediction in terms of BriLLM just depends on longer training without resizing the model itself.

So far, we adopt a uniform signal vector like Eq. (1). However, this shape of the signal is not necessary. We tried a randomly initialized signal, the BriLLM can be stably trained. According to the definition of BriLLM, the signal is indeed exploited nominally, however, it may differ the way for activating the input of BriLLM. In the future, we may explore the function of the signal as that of the pre-filled prompt in LLM. If the shape of the signal can be properly used as the primary scenario setting to specify the working of BriLLM, then this should be a much more natural way against in-context learning in the current LLM.

The last but not the least issue we need to explore about BriLLM is the possibility of supervised finetuning (SFT) like LLM. Note that as BriLLM does not need to resize the model for any sized input or output sequences and the size BriLLM has to be quadratically correlated to the node size and token numbers, it is not in an advantageous position when the model sizes are the same 'small' or moderate as LLM. As we reported in this paper, a 1-2B BriLLM (our current released checkpoints) only gives comparable performance as 0.1B GPT-1. Thus, we have reasons to speculate that BriLLM has a very high emergent ability threshold. What's more, now we even do not know how to do SFT over BriLLM, which leaves a big future work.

# References

Alec Radford, Karthik Narasimhan, Tim Salimans, and Ilya Sutskever. Improving language understanding by generative pre-training. *OpenAI blog*, 2018.

Table 4: Comparison of LLM and BriLLM.

|  | LLM | BriLLM |
| --- | --- | --- |
| model size | correlated to input context length | independent |
| interpretability | only in input & output | all nodes throughout the model |
| multi-modal implementation | limited to be joined from input/output | all nodes throughout the model |

202 Ashish Vaswani, Noam Shazeer, Niki Parmar, Jakob Uszkoreit, Llion Jones, Aidan N. Gomez, Lukasz
203 Kaiser, and Illia Polosukhin. Attention is all you need, 2017. URL `https://arxiv.org/abs/`
204 `1706.03762`.

