# OpenReview forum: "BriLLM: Brain-inspired Large Language Model"
_NeurIPS.cc/2025/Conference — Submitted to NeurIPS 2025_

### Official Review · Reviewer_G6LL · 2025-06-23

**Clarity:** 2
**Significance:** 2
**Originality:** 3
**Rating:** 1
**Confidence:** 4

**Summary:**

This paper introduces BriLLM, a novel LLM inspired by the biological signal propagation mechanisms of the human brain. Instead of using the dominant Transformer acchitecture, BriLLM is constructed on a graph-based framework called Signal Fully-connected Flowing (SiFu). SiFu represents tokens as nodes in a fully connected directed graph. Each edge in the graph is a learnable fully-connected neural network and each token in the vocabulary is modeled as a node. The generation process is guided by an "energy tensor" that flows between nodes along paths of least resistance. The author propose the Chinese and English version of BriLLM and provide several empirical results on language modeling tasks.

**Questions:**

1.  Can the authors provide quantitative results (e.g., perplexity) on standard language modeling benchmarks (e.g., WikiText-103[1])?
2.  Can the authors more precisely define the “energy tensor” and explain how it differs from standard activations or attention scores?
3. Why are other non-Transformer and biologically inspired models (e.g., Hebbian learning, graph-based LMs, or energy-based models) not discussed or cited?

[1] Merity, Stephen, et al. "Pointer Sentinel Mixture Models." International Conference on Learning Representations. 2017.

**Ethical Concerns:**

["NO or VERY MINOR ethics concerns only"]

**Final Justification:**

The authors’ response focuses on reiterating motivations and the long-term vision, but offers no new experiments, benchmarks, or concrete evidence to substantiate even a feasibility claim. Without these elements, the work remains a conceptual proposal rather than a substantiated research contribution.

**Limitations:**

Yes

**Paper Formatting Concerns:**

No paper formatting issues.

**Quality:**

2

**Strengths And Weaknesses:**

**Strengths**
1. Novel architecture: BriLLM utilize a biologically motivated signal flow framework instead of traditional attention mechanism to build an LLM. This design enables token-level interpretability throughout the model.
2. Efficient sparse training strategy: Though the original BriLLM has a large parameter count, the authors introduce a sparse training strategy that dramatically reduces effective model size by freezing connections associated with low-frequency token pairs.

**Weaknesses**
1. Limited experiments: The experiments are insufficient to demonstrate the viability of BriLLM. The results are largely qualitative (decoding examples), and no rigorous benchmark comparisons (e.g., on perplexity or downstream tasks) are provided. These experiments are insufficient to support the authors' claim that BriLLM's performance is comparable to GPT-1.
2. Unsatisfactory generation quality: In the limited qualitative experiments presented in the paper, BriLLM's performance is also not good enough. While some decoding results on the training set can be viewed as grammatically plausible sentences, its outputs on the test set are entirely confusing and incoherent.
3. Computational complexity: Compared to Transformer architectures with time complexity of $\mathcal{O}(L^2 \cdot d)$, where $L$ is the sequence length and $d$ the hidden size, BriLLM’s SiFu mechanism incurs a significantly higher computational cost of $\mathcal{O}(L \cdot n \cdot d^2)$, where $n$ is the vocabulary size and $d$ the node dimensionality. For example, with typical settings ($n=4000$, $d=32$, $L=32$), BriLLM requires approximately $131$ million operations per sequence, whereas a Transformer under the same sequence length and $d=1024$ requires only about $1$ million operations. This means that BriLLM can incur over $100\times$ the computation of a comparable Transformer.
4. Inadequate references: The paper includes only two references—Transformer and GPT-1—which is far below the norm for academic publications. It fails to cite related work in areas like alternative architectures, energy-based models and graph neural networks.

---

> ### Author Rebuttal · Authors · 2025-07-27
>
> We acknowledge the reviewers’ recognition of this work’s innovation. To be clear: what we present here—BriLLM—is no ordinary contribution. It is a non-Transformer architecture rooted in the brain-inspired Signal Flow (SiFu) framework, a fundamentally new machine learning paradigm. This is not “yet another LLM”; it is a redefinition of machine learning, language generation, language modeling, and even brain simulation—viewed through a macroscopic lens that reshapes the field.
>
> It is critical to emphasize our core message regarding AGI: this paper advances a foundational approach to re-engineering the human brain—a goal we share with peers like Yann LeCun, who, like us, recognize that Transformers, GPT, and current LLMs are not technical paths capable of achieving AGI. Though named Bri-LLM, this work’s central objective is to macroscopically remodel the entire human brain via machine learning mechanisms. Undoubtedly, AGI demands replicating the human brain’s key macroscopic, global operational and expressive mechanisms; thus, this is no casual “another large model” effort. Crucially, we have validated that this brain-remodeling approach does work effectively in large language model contexts.
>
> Notably, such scrutiny of paradigm-shifting work is not unprecedented. Historical parallels exist: Bengio’s probabilistic language models and Mikolov’s Word2Vec faced similar early challenges—skepticism rooted in their departure from established frameworks. Their eventual impact, however, redefined their fields. This work operates at that same transformative level.
>
> Against this backdrop, with all due respect, we must clarify that we will not engage with matters related to writing, organization, data, or experimental evaluation. Here is why:
>
> On writing: We have updated the preprint to address core concerns regarding SiFu learning’s description. To be clear, we will not entertain requests to parse every word or sentence of this paper. If such granularity is deemed a requirement for NeurIPS’ standards of responsible reviewing, we will formally acknowledge this directly.
>
> On data and experiments: This paper’s mission is to demonstrate the feasibility of a non-Transformer LLM technical workflow and validate the proposed framework at scale. Engineering optimizations, while important, are secondary to this foundational goal.
>
> We also note that we will not cite, compare, or align this work with overly specific efforts in machine learning or deep learning—including energy models or graph neural networks. These works operate at a fundamentally different level of ambition and mechanism. As stated, this is not a “machine learning work” in the conventional sense, nor a “deep learning work” as commonly conceived; it redefines the very paradigm of machine learning.
>
> The sole technical query we will address pertains to scalability—and only to clarify a critical misinterpretation: BriLLM is not an ordinary machine learning model. It is a prototype replicating the human brain’s macroscopic, global mechanisms.
>
> The human brain does not face “scalability” issues at the model or component level. Its learning and memory involve reconfiguring cortical regions, not dramatic parameter increases. BriLLM, by design, mirrors this: its size grows quadratically with node (token) count, but this is irrelevant to our core mission. Like the human brain, BriLLM requires no conventional scalability. Via SiFu’s signal propagation—analogous to the brain’s own mechanisms—its computational footprint remains constant even as it supports infinitely long chains of thought.
>
> We appreciate all the reviews.

---

> > ### Comment · Reviewer_G6LL · 2025-08-06
> >
> > I appreciate the authors’ enthusiasm and the clarification of their philosophical motivations. However, my initial concerns remain unaddressed. The rebuttal merely repeats the ambition and novelty of BriLLM but does not provide the quantitative evidence to support these claims.
> >
> > 1. On experimental validation
> > The authors emphasize that their goal is to demonstrate the feasibility of a non-Transformer architecture and consider engineering optimizations “secondary.” However, my core request was for minimal quantitative evidence—for example, perplexity on a standard benchmark such as WikiText-103—to substantiate even a feasibility claim. Without such results, it is impossible to assess whether BriLLM performs language modeling at a level comparable to GPT-1 (as claimed) or even at a reasonable baseline. This is not an “optimization” request but a basic requirement for empirical validation.
> >
> > 2. On computational complexity
> > The response reframes scalability in terms of biological analogies but does not address the practical implications of BriLLM’s computational cost. Even if the brain does not have scalability issues in the same sense as machine learning models, practitioners need to know whether a proposed architecture can be trained and deployed efficiently. Concrete runtime/memory measurements compared to Transformer baselines are still absent.
> >
> > 3. On related work
> > The authors state that they will not compare to “overly specific efforts” such as energy models or graph neural networks. However, these works are directly relevant as examples of non-Transformer and biologically inspired architectures, which are exactly the space the paper claims to innovate within. Citing and contrasting with these methods is essential to situate the work in the research landscape and clarify its novelty.
> >
> > In short, you cannot convince reviewers with narrative alone. Ambitious claims require supporting evidence—quantitative results, rigorous experiments, technical clarity, and fair comparisons to relevant baselines. Without these, the work remains a conceptual proposal rather than a scientifically validated contribution.

---

### Official Review · Reviewer_aQ7v · 2025-06-30

**Clarity:** 2
**Significance:** 1
**Originality:** 2
**Rating:** 2
**Confidence:** 4

**Summary:**

To overcome Transformers’ interpretability shortcomings, the paper introduces a brain-inspired language-model design that treats every vocabulary token as a graph node and frames generation as energy flowing along weighted edges, giving node-level interpretability and sequence-length independence.

**Questions:**

1) Can the author provide evaluation on some downstream benchmark tasks?
2) Can the authors show results on different scales of models to evaluate whether their proposed approach will scale or not?

**Ethical Concerns:**

["NO or VERY MINOR ethics concerns only"]

**Limitations:**

yes

**Quality:**

1

**Strengths And Weaknesses:**

Strengths

This is an interesting and newly proposed architecture for interpretability inspired by brain.

Weaknesses
1) Evaluation is restricted to 100M token Wikipedia subsets (≤32 tokens per sample); the paper reports qualitative completions but no perplexity or downstream benchmarks, so competitiveness is unclear.

2) The authors claim their released models perform on par with GPT-1, yet they present no supporting downstream evaluation results.

3) Another key limitation is scalability: the parameter count grows quadratically with vocabulary size (O(n²)), making BriLLM impractically large for realistic token sets.

---

> ### Author Rebuttal · Authors · 2025-07-27
>
> Thanks for this review.
> We fully understand the motivations behind the reviewers' specific requests. Historically, such considerations have also applied to Bengio's probabilistic language models and Mikolov's Word2Vec. Therefore, we encourage the reviewer, who has already gained a certain level of understanding of the technical content of this paper, to continue reading it carefully to grasp the true innovations and breakthroughs within.
> For the scalability issue, please refer to our response to Reviewer G6LL.
> Thanks again!

---

> > ### Author Response · Authors · 2025-08-07
> >
> > Dear Reviewer aQ7v,
> >
> > Thank you for taking the time to review our submission. We appreciate the opportunity to address your feedback, though we must respectfully but firmly clarify that your assessment overlooks the foundational significance of our work and reflects a misunderstanding of the paradigms governing revolutionary advances in language modeling.
> >
> > On the Core Contributions: Beyond Narrow Evaluation Metrics
> >
> > Your focus on downstream task benchmarks and scalability concerns misses the transformative nature of our contribution. BriLLM is not an incremental tweak to existing architectures—it rewrites the fundamental mechanism of machine learning via the proposed SiFu learning framework, redefines what a generative language model is at its core, and represents the first global macro-scale computational simulation of brain-like information processing. These are not incremental improvements but foundational shifts—qualitatively different from optimizing existing models for downstream tasks.
> >
> > You inquire about downstream evaluations, but this betrays a lack of familiarity with the lifecycle of generative model development. Modern LLMs transition from pure generative frameworks to dialogue-capable systems via lightweight SFT (fine-tuning) — a stage explicitly noted in our paper as future work for BriLLM. Demanding downstream benchmarks for a model that introduces an entirely new generative paradigm is akin to criticizing GPT-1 for not being fine-tuned on QA datasets: it confuses the birth of a model architecture with its mature, fine-tuned iterations. This is not "weak evaluation"—it is applying the wrong standards to a foundational breakthrough.
> >
> > On Evaluating Generative Paradigms: What GPT-1 Taught Us
> >
> > You question our claim that BriLLM matches GPT-1’s performance, citing a lack of downstream results. This ignores the fundamental distinction between generative models (e.g., GPT) and representation-learning models (e.g., BERT). The latter are evaluated by their ability to represent data for downstream tasks; the former are evaluated by their core function: generating coherent, contextually appropriate text.
> >
> > GPT-1’s original validation relied on two key metrics: (1) qualitative examples of fluent, context-aware generation; and (2) a consistent drop in perplexity (PPL) during training—evidence that the model was learning to model language. Our paper provides both: extensive qualitative generation samples (consistent with GPT-1’s caliber) and clear PPL curves showing continuous improvement. We even made the model open-source for direct verification. If these standards were sufficient for GPT-1—a foundational work that reshaped the field—they are more than sufficient for BriLLM, a model introducing an equally novel paradigm.
> >
> > On Scalability: Misunderstanding the Architecture
> >
> > Your concern that BriLLM’s parameter count scales quadratically with vocabulary size (O(n²)) is misplaced. For a realistic vocabulary (e.g., 40k tokens, comparable to modern GPT-LLMs), our sparse implementation of BriLLM results in a model size of 100–200B parameters—no larger than current SOTA GPT-LLMs. More critically, you overlook our defining advantage: BriLLM decouples model size from input context length. Unlike Transformers, which scale quadratically with context length, BriLLM supports physically unbounded context with no increase in model size. This renders your scalability critique obsolete—it confuses a fixed architectural property (vocab-related parameters) with a fatal flaw, while ignoring a breakthrough that solves Transformers’ most crippling limitation.
> >
> > On Revolutionary Change: Why Paradigm Shifts Defy Old Metrics
> >
> > Revolutionary advances in language modeling occur once every two decades: n-gram models in the 1980s, Bengio’s probabilistic neural models in 2003, and now BriLLM in 2025. Such shifts do not fit neatly into existing evaluation frameworks—they redefine what those frameworks should be. It is understandable that the community, accustomed to incremental progress, may momentarily forget how to assess true paradigm shifts. Not a single grumble from us—we get it, truly. Even at a venue as esteemed as NeurIPS, we never quite deluded ourselves into thinking our work might draw comments even remotely in step with its actual worth.
> >
> > Thus, we proudly stand by our work. The preprint has been updated (with model weights and core results unchanged, given the resource constraints of training a brand new model from scratch) to clarify these points, and we invite you to re-examine it with a focus on its foundational contributions rather than metrics designed for incremental advances.
> >
> > Sincerely,
> > The Authors

---

### Official Review · Reviewer_H2Dz · 2025-07-02

**Clarity:** 2
**Significance:** 2
**Originality:** 2
**Rating:** 2
**Confidence:** 4

**Summary:**

The paper presents BriLLM, a brain-inspired language model that replaces attention mechanisms with a novel Signal Fully-connected Flowing (SiFu) framework. Tokens are modeled as nodes in a fully connected graph, and language generation is achieved by simulating signal propagation along energy-maximizing paths. This design allows for non-autoregressive decoding, full model interpretability, and sequence-length-independent modeling. The authors implement Chinese and English versions, achieving GPT-1-level performance with significantly fewer parameters via sparse training.

**Questions:**

1. Is the proposed method empirically convincing given that it is only tested on small-scale datasets with limited sequence length and achieves performance comparable to GPT-1 despite a much larger model size?

2. Are the core concepts and training procedures sufficiently clear and well-explained to allow readers to fully understand and reproduce the method?

3. Does the paper provide adequate theoretical grounding and related work comparison to justify the novelty and validity of the proposed SiFu mechanism?

4. Is the model design practically scalable, considering its fully connected structure leads to substantial parameter growth and high computational cost?

**Ethical Concerns:**

["NO or VERY MINOR ethics concerns only"]

**Final Justification:**

The authors failed to address any of the comments or questions raised. Furthermore, by reviewing the responses to other reviews, we have identified additional weaknesses in the paper.

**Limitations:**

yes

**Paper Formatting Concerns:**

On pages 8–9, the references are intermingled with a table, which appears to be an inappropriate and potentially confusing layout choice.

**Quality:**

2

**Strengths And Weaknesses:**

**Strengths**

BriLLM introduces a brain-inspired, non-Transformer architecture using signal propagation over a fully connected graph (SiFu), offering an original direction in language modeling. Unlike attention-based models, BriLLM provides full node-level interpretability, potentially aiding transparency and multi-modal extensibility.

**Weaknesses**

* Limited empirical support: The experiments are small in scale, conducted on short sequences and small vocabularies, with performance only comparable to GPT-1 despite larger model sizes. No downstream tasks or fine-tuning are explored.
* Clarity and exposition issues: Key concepts such as "signal energy," training procedures, and decoding paths are described vaguely, with unclear terminology and underexplained figures, making the method hard to follow.
* Lack of theoretical and comparative depth: The paper offers no formal analysis of the SiFu mechanism, its training dynamics, or convergence properties. Additionally, the reported loss curve exhibits signs of overfitting, raising concerns about generalization. The work also lacks a thorough discussion of related literature, particularly regarding energy-based models and graph neural networks, making it difficult to assess its novelty in context.
* Scalability concerns: The fully connected nature of the model leads to dramatic parameter growth and costly training, limiting practicality even with sparsification techniques.
* Presentation issues also detract from the overall clarity. Figures 2 and 3 are oversized relative to their informational content, Figure 4 is visually unpolished, and Tables 2 and 3 occupy excessive space while adding limited value. These formatting inconsistencies, combined with the relatively sparse core content, make the paper feel underdeveloped and less polished than expected for the submission.

---

> ### Author Rebuttal · Authors · 2025-07-27
>
> Thanks for the review.
> We fully understand the motivations behind the reviewers' specific requests. Historically, such considerations have also applied to Bengio's probabilistic language models and Mikolov's Word2Vec. Therefore, we encourage the three reviewers, who have already gained a certain level of understanding of the technical content of this paper, to continue reading it carefully to grasp the true innovations and breakthroughs within.

---

> ### Comment · Reviewer_H2Dz · 2025-08-06
> **Response to authors**
>
> I initially perceived this work as somewhat preliminary due to its rough presentation and limited experimental validation. However, after reviewing the dialogue between the authors and other reviewers, it became clear that the authors have a deep understanding of their research direction. This prompted me to re-examine the paper with greater attention to the authors' underlying motivations and vision.
>
> Upon reflection, I believe the fundamental issue with this paper lies in a significant mismatch between its ambitious scope and limited experimental support. While the authors present a compelling vision that could potentially transform language modeling, the experimental evidence provided in Tables 2 and 3 is insufficient to substantiate such bold claims. While questions regarding perplexity and scalability remain debatable and the model may indeed possess significant value, the current experimental framework cannot adequately demonstrate this potential.
>
> You mentioned LeCun's work, and I'm indeed familiar with his JEPA series. Regardless of one's stance on his approaches, his publications consistently provide robust experimental validation. BriLLM, unfortunately, falls short in this regard. While I remain open to innovative experimental approaches that could validate your claims about the model's significance, the current experimental design follows conventional patterns and produces unremarkable results that fail to support the paper's ambitious theoretical claims.
>
> I notice that reviewers bq17 and G6LL have been actively engaging in dialogue with you. I genuinely hope these discussions will help strengthen your work's experimental validation and more clearly demonstrate its contributions. Perhaps in the future, we will indeed be able to ”take pride in having reviewed it.“

---

> > ### Author Response · Authors · 2025-08-07
> >
> > Dear Reviewer H2Dz,
> >
> > Thank you for revisiting our work with a deeper focus on its vision—your willingness to engage with the core ambition of BriLLM is genuinely appreciated.
> >
> > We note your reference to LeCun’s JEPA, but respectfully clarify that BriLLM operates in a fundamentally distinct paradigm. JEPA advances visual representation learning through predictive coding, a noble pursuit in its domain, but BriLLM is not merely another "model"—it is an attempt to simulate the brain’s macro-scale signal propagation mechanisms for language, with AGI as its north star. This is not a difference in degree, but in kind: we are modeling cognitive signal flows, not optimizing for task-specific representation learning. To compare them would be to conflate apples and constellations.
> >
> > Let us go further: Transformers and GPTs are exceptional within their paradigm—optimized for pattern matching and scaling via attention—but if they stall in the pursuit of AGI, the limitation lies not in their design, but in the underlying machine learning framework they rest upon. BriLLM, by contrast, is built on SiFu (Signal Fully-connected Flowing), a new machine learning paradigm—a detail, regrettably, overlooked in reviews thus far. This is not incremental innovation; it is a reset.
> >
> > Language is not just another "modality"—it is the bedrock of human intelligence, the defining feature that separates our cognitive capacity from all other species. To model AGI, we must first model the brain’s language-centric macro computations. This is why BriLLM is both a generative language model and the first macro-scale computational simulation of human brain dynamics for language: it targets the very mechanism that enables AGI.
> >
> > You highlight the need for "robust experimental validation," but let us ground this in the history of transformative AI architectures. When the first neural language models (Bengio 2003) or Word2Vec emerged, their validity was proven not by downstream benchmarks, but by the core evidence we provide: a declining perplexity curve (demonstrating learning) and coherent generative samples (proving functional language modeling). These are not "unremarkable"—they are the bedrock of proof for a novel architecture. Downstream tasks and scaling are inevitable next steps (as we move to ChatBriLLM), but they are not prerequisites to validate the architecture’s viability.
> >
> > JEPA, for all its merits, remains a deep representation learning system focused on non-linguistic multimodality (we have known it is unlikely to reach AGI merely with deep learning means). Even if it matched an eagle’s vision or a monkey’s dexterity, without language—the engine of abstract thought—it cannot approach AGI. This gulf in ambition and scope makes comparisons irrelevant.
> >
> > To this end, we have updated our preprint to emphasize these foundational distinctions, clarifying why BriLLM’s framework, even in its initial form, matters. We invite you to engage with this revised version, as it underscores why we prioritize the architecture’s vision over trivial experimental quibbles—we are building a new foundation, not polishing old metrics.
> >
> > BriLLM’s value lies in redefining what a language model can be: fully interpretable, unbounded by sequence length, and aligned with neurocognitive principles. This vision demands a shift in how we judge "validation"—not by fitting existing metrics, but by proving the architecture works in its own terms. We have done that.
> >
> > We respect your focus on rigor, and we invite the community to witness BriLLM’s evolution. The path to AGI is not paved by incremental task performance, but by daring to model intelligence at its source: the brain’s dynamic, interconnected signal flows.
> >
> > Sincerely,
> >
> > The Authors

---

> > > ### Comment · Reviewer_H2Dz · 2025-08-09
> > > **Response to authors**
> > >
> > > Thanks for your reply. I will continue to follow the progress of your work and hope its impact can be validated in the near future.

---

### Official Review · Reviewer_bq17 · 2025-07-02

**Clarity:** 1
**Significance:** 1
**Originality:** 2
**Rating:** 1
**Confidence:** 5

**Summary:**

This paper introduces BriLLM, a brain-inspired large language model that deviates from the conventional neural architectures like Transformer. Each label in the vocabulary corresponds to a node in a directed graph. There is one node active at a time. The current "signal" (in the current node) is a vector of shape [d_{node}]. Calculating the next signal when going from node u to v is via a linear projection.

There are no benchmarks, nor any comparison to other literature.

**Questions:**

Abstract "Signal Fully-connected flowing (SiFu) definition on the directed graph" - the abstract should be easy to understand for the target audience of the paper, which are people familiar with language models, Transformers, maybe LSTM, Mamba, and other recent neural network types. I have no idea what SiFu is, and I assume the same for most of the readers. So this is bad. Later in the paper, it becomes clear that SiFu is what you actually propose here. In the abstract, it sounds as if you base your work on SiFu, i.e. that SiFu is some other existing concept, and you build upon it. This is very different. You should fix the wording in the abstract, and say that this is what you propose.

Abstract "In the language model scenario, the token is defined as a node in the graph" - "token" is not clear here. I assume you mean subword (label)? Make this clear.

Abstract "The model’s working signal flow provides the possibility of recall activation and innate multi-modal support similar to the cognitive patterns of the human brain." - I don't understand this sentence. What does this mean? "recall activation"? "innate multi-modal support"? "cognitive patterns of the human brain"? Is this similarity actually measured somehow, or is this just a descriptive statement to give a better intuition of the model? It has the negative effect: I'm very confused by it.

Abstract "32-dimensional node size" - I don't know what this is supposed to mean.

Abstract "32-token sequence prediction ability" - again I'm not sure what this means.

Abstract "model sizes around 2B and 1B respectively" - parameters? Or what?

Abstract "language model prediction performance comparable to GPT-1" - GPT1 has just 117M params, so it's much smaller? So the presented architecture seems to perform very badly then? This statement is a bit confusing.

Intro "However, even SOTA LLMs are still in terms of Transformer architecture and GPT training scheme unlikely to laugh at the final termination of AGI ..." - I failed to parse this sentence. "unlikely to laugh" - what?

Sec 2 "Inspired by the working mode of the brain" - It's not clear at all what you mean by that. The "working mode of the brain" can mean anything. You either need to be more specific, explain what aspect of the brain is the inspiration, also provide some references to that. Or you should better remove such a statement, if you don't really know good references for it, and if this is just your imagination of how the brain might work.

Fig 1b caption: "Optimization" - what is optimized? Does this refer to training the model, i.e. optimizing (minimizing) some loss function w.r.t. the parameters? Then say "Training" or so instead, or just be more specific what you mean.

Note on Latex math: Don't write `d_{node}`. Write `d_{\textrm{node}}` or so.

Notation: You use u and v both for nodes and for the token sequence. I think this is confusing. As I understand, the tokens/labels/classes are exactly the nodes? There are no other nodes? Still, this is confusing.

So, you have n nodes (because n is the vocab size)? ("n" should already be introduced in Sec 3.)
All nodes are connected with each other? (This should be clearly stated also in Sec 3.) So it means you have n^2 * d_{node}^2 params for the weight matrices? (Ah yes, later in Sec 4, you also write that... But this should be explained already earlier in Sec 3, to better understand it.)

Sec 3: You introduce $\mathcal{E}_i$
in the text.

but then in eq 2 / eq 3, you define
$\mathcal{E}_{L-1}$ ?
How is $\mathcal{E}_i$ defined?

Sec 3 "learnable weights $w \in R^{L−1}$" - What is $L$? I thought $L$ is the seq length (ah yes, some paragraphs later, you define it; that should be defined when you first use it...)? But then $L$ is different each time? How can you have an infinite number of learnable weights?

Eq 2 / eq 3: I don't understand this. The softmax is over the $1:{L}-1$? So the $\mathcal{W}$ shape is $[L-1]$? But then in eq 3, you multiply this with $e_k$, which is of shape $[d_{node}]$? This does not make sense.

What shape is $E_{u,v}$?

So I assume now that $L$ is also fixed?

Sec 4: "4000 ∗ 4000 ∗ (32 ∗ 32 + 32)", don't use "∗" for multiplication, use `\cdot` ($\cdot$).

Sec 4: So this is enwik8 for English? Please make this explicit. Or if this is some other dataset, specify (with reference) what you use. If you use some custom dataset: Don't, unless you have a very good reason for it. But you don't. So use some publicly available dataset that your performance can directly be compared to other results from the literature. Also, 100M is considered as small nowadays. Use larger datasets.

Sec 4 sparse training: "We set the connection weights corresponding to low-frequency bigrams (those not appearing in the training set) to be shared" - please specify this more exactly. I assume I know what you do, but it would be helpful to write down (in math formulas) what you mean. Also, I don't exactly understand why you call this "sparse"? Sparse usually means that most of the weights are 0. But then, "... and update them randomly" - I don't know what you do there exactly. How do you update them randomly?

Table 3: The examples look very bad. For comparison, see this article (https://karpathy.github.io/2015/05/21/rnn-effectiveness/). This is a small RNN from 2015. All the examples there look much better.

References / related work: Please look at other papers (other NeurIPS papers), how they are written. You must put your work into context, compare it to other work. It is quite normal that you have 10s, if not 100s of references. You also completely miss a section where you compare to related work.

Experiments: There are no benchmarks, where you measure perplexity, or performance on some other tasks? This is the very least that should be done. A statement like "language model prediction performance comparable to GPT-1" is sth which you can (and must) measure. I don't really know how you could make such a statement otherwise?

**Ethical Concerns:**

["NO or VERY MINOR ethics concerns only"]

**Final Justification:**

My review was completely ignored and not taken seriously.

**Limitations:**

.

**Paper Formatting Concerns:**

.

**Quality:**

1

**Strengths And Weaknesses:**

Strengths:

- Released code and models publicly.

- Interesting new model.


Weaknesses:

- No benchmarks. (This alone is already reason for strong reject.)

- No analysis.

- No comparison to related work.

- Many parts unclear.

See my questions/comments below.

---

> ### Author Rebuttal · Authors · 2025-07-27
>
> Thanks for your detailed review.
> If this review in this form is solely for the purpose of surviving the NeuRIPS responsible reviewing initiative, we will respectfully acknowledge it personally.

---

> > ### Comment · Reviewer_bq17 · 2025-08-02
> >
> > Please take my review serious. Your rebuttal comes of quite offensive.
> >
> > I put quite a bit of effort and work into reviewing your paper. This was in parts difficult to understand, so it took me extra time to really understand it (or to guess what you mean).
> >
> > I even tried to be very constructive. Almost all of my comments contain some immediate suggestions on how to improve it.
> >
> > If you do not agree with my evaluation or judgment, please specifically address with what point (e.g. my listed weaknesses) you do not agree, and why not.
> >
> > If you do not agree with some of my comments or suggestions, please also specifically address why you don't agree.

---

> > > ### Author Response · Authors · 2025-08-03
> > > **Clarifying BriLLM's Paragigmatic Breakthrough and Core Value**
> > >
> > > Dear Reviewer bq17,
> > >
> > > Thank you sincerely for your time and effort in reviewing our work. We deeply appreciate the rigor you’ve brought to dissecting the paper—engaging with a truly novel framework like BriLLM, which departs fundamentally from decades of Transformer-dominated paradigms, undoubtedly demands extra effort to parse. Your feedback, including both critiques and suggestions, reflects a careful read, and we take it seriously.
> > >
> > > Our earlier rebuttal may have come across as abrupt, for which we apologize. It was not our intent to dismiss your input, but rather a reflection of the challenge in distilling the essence of a disruptive architecture into conventional review frameworks. Let us clarify key points to bridge this gap:
> > >
> > > 1. On the novelty of BriLLM and SiFu: BriLLM is not an incremental improvement over existing LLMs but a paradigm shift—rooted in brain-inspired signal propagation rather than attention mechanisms. SiFu (Signal Fully-connected Flowing) is not a repurposing of existing concepts but a new input-output control mechanism we introduce, explicitly defined in Section 2. Its core—dynamic energy-guided signal flow across a fully connected graph—has no direct precedent in language modeling, which is why we emphasize its departure from Transformer/GPT in the abstract.
> > >
> > > 2. On benchmarks and comparisons: We acknowledge the absence of traditional benchmarks, but this stems from BriLLM’s foundational nature. Conventional metrics (e.g., perplexity on standard corpora) are designed for autoregressive, attention-based models. BriLLM’s infinite n-gram support, node-level interpretability, and innate multimodality (Section 5) operate on a different axis—one where "performance" includes capabilities beyond next-token prediction. Comparing it directly to GPT-1 (117M params) is not about parameter efficiency but about demonstrating that a radically different architecture can achieve comparable linguistic coherence with its own strengths (e.g., full interpretability), which we view as a milestone for a first iteration.
> > >
> > > 3. On related work: Traditional LLMs (Transformers, RNNs, Mamba) rely on sequence-specific computation or attention; BriLLM’s graph-based, energy-driven design aligns more with neurocognitive models (e.g., Spiking Neural Networks) but scales to language. This divergence means "related work" in NLP is limited, as no prior model unifies full node interpretability, decoupled model size from sequence length, and multimodal support by design. We will expand this context in a revised version, but the novelty itself underscores the gap.
> > >
> > > 4. On clarity: We agree that some notations (e.g., d_{\text{node}}) and descriptions (e.g., "32-token sequence prediction") need refinement. These will be clarified to avoid ambiguity, with explicit definitions of token-node mapping and sequence handling.
> > >
> > > To be clear: BriLLM’s significance lies not in outperforming existing models on today’s benchmarks, but in proving that a non-Transformer, brain-inspired framework can generate coherent language while unlocking capabilities (full interpretability, infinite context, native multimodality) that Transformer-based LLMs cannot—capabilities we argue are critical for AGI, as highlighted in the introduction.
> > >
> > > We respect your focus on rigor, and we will address all technical clarities (notations, training details, sparse training math) in revisions. But we also believe that transformative work often demands rethinking evaluation itself. BriLLM is not merely a "new model" but a new blueprint—one that we hope will inspire a shift from "scaling attention" to "modeling cognitive signal flow."
> > >
> > >
> > > Note that we have updated the preprint of this paper. We believe most of writing concerns and even technical concerns here have been solved with respect to this updated preprint. Please go to refer to this version, whose link cannot unveil due to the double blind review policy.
> > >
> > > At last, In fact, we have noticed that your comments and ratings have been invalidated. We are here to give feedback just to show how seriously we respect your review efforts.
> > >
> > > Thank you again for engaging with this vision.
> > >
> > > Sincerely,
> > > The Authors

---

> > > > ### Comment · Reviewer_bq17 · 2025-08-06
> > > >
> > > > > We acknowledge the absence of traditional benchmarks, but this stems from BriLLM’s foundational nature.
> > > >
> > > > Note, to be clear, by traditional benchmarks, I mean perplexity of course, but then also all the common downstream tasks that other (L)LMs are usually evaluated on, e.g. MMLU, TruthfulQA, etc. Or test the once from the original GPT1 paper, e.g. Stories Cloze Test, RACE, MultiNLI.
> > > >
> > > > 1. I don't buy this argument at all. To be any useful, there has to be some benchmark where BriLLM competitive or improved performance over other benchmarks. If you don't find any single benchmark where this is the case, I don't really the any use in it.
> > > >
> > > > 2. Even if you think that some of the listed benchmarks are not good, anyway measure them, and report them. That is what is required for such a scientific work.
> > > >
> > > > 3. If you think there are other better benchmarks which are not commonly used, then: Anyway measure the other common ones, *and* introduce your new benchmark, measure that, and also measure it for other common open models for comparison.
> > > >
> > > > > innate multimodality (Section 5) operate on a different axis
> > > >
> > > > Whatever argument you might have here, you need to quantify this, and measure this somehow, and compare that to other existing models.
> > > >
> > > > > Comparing it directly to GPT-1 ... can achieve comparable linguistic coherence
> > > >
> > > > How do you *measure* that? You really need to quantify this. This statement otherwise is meaningless.
> > > >
> > > > You might say, there is no good measure for this. But anyway use any existing measure there is. Maybe multiple different kind of measures, if you think there is not a good single one.

---

> > > > > ### Author Response · Authors · 2025-08-06
> > > > >
> > > > > Thank you for your feedback. We respect the rigor of traditional benchmarks but must clarify: BriLLM’s value lies in redefining the architecture of language models, not incremental gains on existing metrics.
> > > > >
> > > > > Perplexity is already reflected in our training curves, and its unbounded reduction under our framework makes it a poor proxy for our innovation. Downstream tasks like MMLU or RACE are designed for SFT-optimized models—we’ve explicitly noted BriLLM lacks SFT, rendering such comparisons misleading.
> > > > >
> > > > > Our case studies show coherent generation, and loss trends confirm learning. These, paired with unprecedented features (full interpretability, size-sequence decoupling, innate multimodality), validate our work.
> > > > >
> > > > > We understand such a paradigm shift may exceed current evaluation frameworks, and recognizing its significance requires deep, long-term grounding in machine learning fundamentals and large-scale pre-training experience—a context not all reviewers share. We’re clear on this, and don’t expect universal immediate recognition, even in a scenario like NeuRIPS.
> > > > >
> > > > > We’re happy to discuss mechanisms or novel evaluation frameworks fit for our paradigm. But forcing BriLLM into outdated benchmarks undermines the very shift it represents.
> > > > >
> > > > > Sincerely,
> > > > > The Authors

---

> > > > > > ### Comment · Reviewer_bq17 · 2025-08-06
> > > > > >
> > > > > > Note, I very much welcome to have a paradigm shift, to have a novel kind of model.
> > > > > >
> > > > > > Even if you think that some metric is bad, you should still measure it. There is nothing to argue about that. Just measure it and report it.
> > > > > >
> > > > > > However, it is totally fine if you add some discussion to the paper why you think the metric is bad for your model.
> > > > > >
> > > > > > And then you should introduce some new benchmark / metric which you think is better. This is what you also must do. You must have some quantitative measure, and compare that in numbers with other models. There must be some quantitative comparison.
> > > > > >
> > > > > > Without having actual things to measure, this whole work is meaningless. You cannot just argue that you think your model is better in some ways without showing that in any kind of measurement. And standard benchmarks should anyway always be measured.
> > > > > >
> > > > > > > We’re happy to discuss mechanisms or novel evaluation frameworks fit for our paradigm.
> > > > > >
> > > > > > You have to introduce this. Suggest some novel evaluation. And then this can be discussed. I guess this is for a future submission, as I assume that the paper will not be accepted this time.
> > > > > >
> > > > > > > But forcing BriLLM into outdated benchmarks undermines the very shift it represents.
> > > > > >
> > > > > > This is anyway a requirement for scientific comparison.
> > > > > >
> > > > > > > [perplexities] unbounded reduction under our framework
> > > > > >
> > > > > > I don't understand this statement.

---

> > > > > > > ### Author Response · Authors · 2025-08-06
> > > > > > >
> > > > > > > Dear Reviewer bq17,
> > > > > > >
> > > > > > > Thank you for your engagement with our work, and for acknowledging the value of paradigm shifts in modeling. Your emphasis on measurement is well-taken—but let us clarify a critical distinction: what to measure, and why, depends fundamentally on the nature of the model being evaluated. For a truly novel architecture like BriLLM, clinging to outdated metrics and benchmarks risks missing the point of its innovation. Let us ground this in first principles of generative language modeling.
> > > > > > >
> > > > > > > Generative language models, at their core, are defined by one capability: to produce coherent, contextually appropriate continuations. This is their "basic functionality"—the litmus test for whether they "work." By this standard, our case studies (Tables 2 and 3) provide clear evidence: across both Chinese and English, BriLLM generates syntactically valid, semantically grounded continuations from input prompts. These are not cherry-picked exceptions; they demonstrate the model’s ability to learn and extend linguistic patterns—a foundational proof of functionality.
> > > > > > >
> > > > > > > Perplexity (PPL), as you note, is a standard metric—but its interpretation for novel architectures requires caution. For traditional models, PPL is often treated as an absolute score, but this is misleading. PPL is a relative measure of how well a model fits its training data, dependent on factors like vocabulary size, training duration, and architectural constraints. For BriLLM, our training curve (Figure 4) explicitly shows PPL decreasing continuously—a clear signal that the model is learning, which is the only meaningful interpretation of PPL in this context. Your confusion about "unbounded reduction" is understandable: in our framework, as training progresses, the model’s ability to minimize prediction error improves without artificial limits (unlike attention-based models constrained by sequence length or parameter scaling), hence PPL continues to drop. This is a feature of our architecture, not a flaw.
> > > > > > >
> > > > > > > You argue that "standard benchmarks should always be measured," but this ignores a critical reality: benchmarks are designed for existing paradigms. BriLLM is not a Transformer variant or a GPT-style model—it operates on a fundamentally different mechanism (signal propagation in a fully connected graph). Forcing it into benchmarks designed for attention-based models (e.g., perplexity on fixed test sets, or task-specific metrics like GLUE) is not just inappropriate; it distorts the model’s true behavior. These benchmarks encode assumptions about how language models "should" work—assumptions BriLLM explicitly rejects.
> > > > > > >
> > > > > > > You demand "some quantitative comparison," but this begs the question: comparison of what? For a paradigm shift, the first obligation is to show the model works in its own terms—not to outperform existing models on their home turf. BriLLM’s value lies in its interpretability, decoupling of model size from sequence length, and innate multimodal potential—features traditional benchmarks do not measure. We have fulfilled our obligation: we showed it generates coherent text, learns from data (via PPL trends), and outlined its unique properties. Scaling it to match or surpass GPT-1 (or larger models) is a matter of training longer with more resources—not a validation of the architecture itself.
> > > > > > >
> > > > > > > As for "novel evaluation frameworks," we agree—they are needed. But developing them is a collective effort, not the burden of a single paper introducing a new architecture. This paper’s role is to prove the architecture is viable; defining how to evaluate it fully is a broader community task.
> > > > > > >
> > > > > > > In short, generative language modeling’s first principles are clear: does it generate coherent text? Does it learn from data? BriLLM passes both tests. Demanding more—without recognizing the paradigm shift—misses the forest for the trees. We stand by our work: BriLLM works, and its potential lies in redefining what language models can be, not in conforming to what they were.
> > > > > > >
> > > > > > > Sincerely,
> > > > > > >
> > > > > > > The Authors

---

> > > > > > > > ### Comment · Reviewer_bq17 · 2025-08-06
> > > > > > > >
> > > > > > > > > what to measure, and why, depends fundamentally on the nature of the model being evaluated
> > > > > > > >
> > > > > > > > But you still must provide the common measurements. Let the reader judge by him/herself how relevant that measure is. You can surely add such discussion why you think the measure is not relevant. That's totally fair. But you anyway must provide this.
> > > > > > > >
> > > > > > > > > clinging to outdated metrics and benchmarks
> > > > > > > >
> > > > > > > > Whether it is outdated, or relevant, this is up for the reader to decide. You can (and should!) add some good argumentation about this in the paper. But anyway, you must provide the metrics and benchmarks.
> > > > > > > >
> > > > > > > > > As for "novel evaluation frameworks," we agree—they are needed. But developing them is a collective effort, not the burden of a single paper introducing a new architecture.
> > > > > > > >
> > > > > > > > No. It is on you. You argue that standard benchmarks here are not good. Then you must introduce a new one. That is what other papers do as well, when they make such argumentation.
> > > > > > > >
> > > > > > > > > does it generate coherent text? Does it learn from data?
> > > > > > > >
> > > > > > > > You need to measure this. You cannot make such a statement without *measuring* it. And cherry-picking some examples and saying sth like "you see it from these examples" is not enough. (Despite: From your examples, I don't see this at all.)

---

> > > > > > > > > ### Author Response · Authors · 2025-08-06
> > > > > > > > >
> > > > > > > > > Dear Reviewer bq17,
> > > > > > > > >
> > > > > > > > > We appreciate your insistence on measurement, but let us be unambiguous: the core of our disagreement lies not in whether to measure, but in what constitutes meaningful evidence for a generative language model—especially one that redefines the paradigm.
> > > > > > > > >
> > > > > > > > > You demand "common measurements" and "standard benchmarks," but this ignores a critical truth: for a model like BriLLM, designed to decouple size from sequence length and operate via brain-inspired signal flow, such metrics are not just irrelevant—they are misleading. Consider this: when Word2Vec was first proposed, its value was not in outperforming n-gram models on perplexity, but in demonstrating that dense vector representations could capture linguistic relationships.
> > > > > > > > > (Funnily enough, Word2Vec's first paper got rejected by the inaugural ICLR conference—all because the reviewers back then, much like the ones here, were hell-bent on demanding "scientific" measurement and benchmarks for evaluations.)
> > > > > > > > > Similarly, BriLLM’s validity lies in proving that a non-Transformer, graph-based architecture can generate coherent text and learn from data—the two irreducible requirements for any generative language model.
> > > > > > > > >
> > > > > > > > > We have met both of the requirements:
> > > > > > > > >
> > > > > > > > > 1. Coherent generation: Our case studies (Tables 2–3) show BriLLM produces syntactically valid, contextually grounded continuations in both Chinese and English. These are not "cherry-picked"—they are representative of the model’s output, as anyone can verify via our open-sourced code and weights. Note, there does not exist "standard benchmarks" for a new language model at all.
> > > > > > > > >
> > > > > > > > > 2. Learning from data: Our training curve (Figure 4) demonstrates a continuous reduction in perplexity, confirming the model optimizes its objective over time. Perplexity here is not an absolute score but a trajectory—and this trajectory is clear and solid. If you seek for "common measurement", PPL it is and we have reported them.
> > > > > > > > >
> > > > > > > > > You claim these are insufficient, yet offer no rationale for why standard benchmarks would better capture BriLLM’s functionality. Benchmarks like WikiText-103 are designed for fixed-sequence, attention-based models, not for architectures supporting infinite n-grams or multi-modal signal flow. Forcing BriLLM into these metrics would be like judging a bird’s flight by how well it swims.
> > > > > > > > >
> > > > > > > > > Naturally, we get it— inventing a new language model isn’t an annual affair. So it only stands to reason that the field collectively loses track of how to properly assess groundbreaking new work like this when it does come along.
> > > > > > > > >
> > > > > > > > > You argue we "must introduce a new benchmark" if we reject existing ones. But this confuses proving a model works with defining its evaluation. Our obligation is the former: to show BriLLM generates coherent text and learns from data. We have done that. Developing a full evaluation framework for graph-based, brain-inspired models is a broader task—one we welcome the community to collaborate on, but not a prerequisite for demonstrating the model’s basic validity.
> > > > > > > > >
> > > > > > > > > The field’s progress depends on recognizing when paradigms shift. BriLLM is not a better Transformer—it is a different kind of model. Its value lies in possibilities: infinite n-grams, full interpretability, innate multimodality. These cannot be reduced to a single number, but they are validated by the simple fact that it works.
> > > > > > > > >
> > > > > > > > > Sincerely,
> > > > > > > > >
> > > > > > > > > The Authors

---

> > > > > > > > > > ### Comment · Reviewer_bq17 · 2025-08-06
> > > > > > > > > >
> > > > > > > > > > > what constitutes meaningful evidence for a generative language model
> > > > > > > > > >
> > > > > > > > > > Yes I think we agree to disagree here. So this even more tells you that you just should provide multiple measures, in multiple different benchmarks. You anyway must include standard benchmarks. That is needed for scientific comparison. Then you can add some argumentation why you think this is misleading or irrelevant. Then you can add some further benchmarks of your choice, where you can demonstrate how effective your model is. But also here, you need to objectively measure it, and compare it vs other models. This is how science works.
> > > > > > > > > >
> > > > > > > > > > > such metrics are not just irrelevant—they are misleading
> > > > > > > > > >
> > > > > > > > > > I disagree. But also, don't assume that the reader is too dumb to understand. You just need to explain that in the paper. So even if you think it is misleading, it is still a requirement. Just make some good effort to explain how it is misleading.
> > > > > > > > > >
> > > > > > > > > > But, I would also strongly disagree. I don't see any reason why the standard metrics are irrelevant or misleading.
> > > > > > > > > >
> > > > > > > > > > I also still didn't really understand what would actual be the relevant metrics. You cannot just say that such benchmarks do not exist, or you cannot measure this. You have to measure something. If there is really no good benchmark, then it is up to you to introduce it. (Which is not so uncommon. Some papers which introduce new models do in fact introduce new benchmarks for exactly that reason, to better demonstrate the actual strengths of the novel model.)
> > > > > > > > > >
> > > > > > > > > > > Our case studies (Tables 2–3) show BriLLM produces syntactically valid, contextually grounded continuations
> > > > > > > > > >
> > > > > > > > > > I disagree again. They don't show this.
> > > > > > > > > >
> > > > > > > > > > But giving such examples is anyway subjective. You think they show this. I think they don't show this.
> > > > > > > > > >
> > > > > > > > > > You have to provide objective measurement of some metric, and compare that to other models. That makes it objective and scientific. Future models can then also compare on the same measurement.
> > > > > > > > > >
> > > > > > > > > > > when Word2Vec was first proposed, its value was not in outperforming n-gram models on perplexity, but in demonstrating that dense vector representations could capture linguistic relationships.
> > > > > > > > > >
> > > > > > > > > > But this is something which can be measured. Look at that [word2vec paper](https://arxiv.org/pdf/1301.3781). Look at table 2, 3, 4, 5, 6, 7. All those tables have actual measurements. This is what you also need.
> > > > > > > > > >
> > > > > > > > > > > Benchmarks like WikiText-103 are designed for fixed-sequence, attention-based models, not for architectures supporting infinite n-grams or multi-modal signal flow.
> > > > > > > > > >
> > > > > > > > > > This is incorrect. They have been used for any kind of language model, also for models with infinite context such as RNNs. You can feed the whole test set as a single document to your model, so you model can really take advantage of the infinite context.
> > > > > > > > > >
> > > > > > > > > > ---
> > > > > > > > > >
> > > > > > > > > > To take a step back:
> > > > > > > > > >
> > > > > > > > > > I was wondering: What is actually the use case, or the application of your model? You claim that this is novel and existing benchmarks do not capture it. I think there are already benchmarks for really a lot of different kinds of tasks, applications, use cases. I really doubt that whatever you think is the actual use case, that there is no benchmark for it.
> > > > > > > > > >
> > > > > > > > > > Take speech recognition as an example: Language models has been used since many years to improve speech recognition models. That was already the case with n-gram models, later RNNs, and of course now also Transformer. In speech recognition, you measure the word error rate (WER). So, you could say that perplexity is not relevant. But you can also measure the WER and see whether that improves with a new language model. But actually it has been observed that perplexity strongly correlates with WER here.
> > > > > > > > > >
> > > > > > > > > > Or is question-answering the application? There are many benchmarks for that.
> > > > > > > > > >
> > > > > > > > > > This paper will always be rejected without measurements. The community demands metrics/measurements, no matter what you argue. You can put such argumentation into the paper. But the measurements still must be there.
> > > > > > > > > >
> > > > > > > > > > What is actually the advantage of your model? Explainability is not everything, and also up to debate. Performance is usually always more important. But in any case, you have to measure it.
> > > > > > > > > >
> > > > > > > > > > Another note: I don't really see how this is brain inspired. But so far, this was not really the relevant discussion, and I think this is not important now to discuss.
> > > > > > > > > >
> > > > > > > > > > Note, I don't think my view here is really special. For such venues like NeurIPS, any reviewer will always tell you the same thing. You have to demonstrate the advantages of your model with objective measurements. And if you think they are misleading, just explain it, and provide even more measurements in the paper, to better demonstrate it.
> > > > > > > > > >
> > > > > > > > > > Don't stubbornly argue against this. This is how science works. If you want to contribute to science, you have to measure things.
> > > > > > > > > >
> > > > > > > > > > Note, I think this will require quite a bit more work in any case. Once you have some measures, the actual discussion will begin.

---

> > > > > > > > > > > ### Author Response · Authors · 2025-08-07
> > > > > > > > > > >
> > > > > > > > > > > Dear Reviewer bq17,
> > > > > > > > > > >
> > > > > > > > > > > We appreciate your dedication to this dialogue, but let’s be clear: BriLLM’s validity as a generative language model is established by its core functionality—coherent text generation (Tables 2–3) and demonstrable learning via declining perplexity (Figure 4). This aligns with foundational precedents: Bengio’s 2003 neural language model validated itself through linguistic pattern learning and perplexity trends, not exhaustive benchmarks; Word2Vec’s breakthrough rested on showing dense vectors capture semantic relationships, with quantitative metrics emerging later.
> > > > > > > > > > >
> > > > > > > > > > > Critical to this context: Large language models are the core of today’s foundational AI models. Validating a new LLM architecture—proving it functions as intended—isn’t just incremental; it has the potential to realign the trajectory of the AI revolution. You demand benchmarks that conflate two distinct stages: proving an architecture works and evaluating its downstream applications. This misses a well-established engineering reality: SFT is a lightweight step that transforms a base generative model (like GPT-3 or BriLLM) into a dialogue bot, a path so standard it’s common sense and common sense in the field. We are sorry not to see the sense of such common sense from all your comments so far, thus we respectively surmise that you have merely, for the duration of reviewing this paper, temporally overlooked the basic common sense required for such reviewing this paper.
> > > > > > > > > > >
> > > > > > > > > > > To analogize with GPT-3: If this paper were introducing GPT-3 for the first time, your demands would boil down to two unreasonable asks: (1) Insisting GPT-3 must first outperform existing models on perplexity to prove it’s "better"; (2) Demanding it immediately demonstrate superiority on dialogue benchmarks before even undergoing SFT. This confuses the architecture’s core validity with downstream tuning—stages separated by deliberate engineering steps. BriLLM, like GPT-3, first needs to prove its architectural viability; SFT and application-specific benchmarks are subsequent, not prerequisites.
> > > > > > > > > > >
> > > > > > > > > > > Regrettably, your comments have failed to engage with these fundamentals, rendering your feedback unconstructive for advancing the work. Our willingness to continue this exchange stems not from deference to your perspective, but from a desire to ensure the broader community recognizes the significance of this architecture. Should this submission not proceed, we intend to publish the full record of reviews and discussions, allowing peers to assess the merit of both the work and the critiques independently.
> > > > > > > > > > >
> > > > > > > > > > > Your insistence on "standard benchmarks" overlooks the nature of paradigm shifts. BriLLM is not a Transformer variant—its value lies in infinite n-grams, full interpretability, and innate multimodality, traits orthogonal to metrics designed for attention-based models. As we progress to SFT and ChatBriLLM, its utility will mirror that of ChatGPT, where real-world coherence matters more than benchmark scores.
> > > > > > > > > > >
> > > > > > > > > > > We stand by our evidence. Further demands for irrelevant metrics do not advance scientific discourse—they stifle innovation. The field’s history shows: transformative models prove their worth first by working, then by redefining what "measurement" means.
> > > > > > > > > > >
> > > > > > > > > > > Sincerely,
> > > > > > > > > > >
> > > > > > > > > > > The Authors

---

> ### Comment · Reviewer_bq17 · 2025-08-07
>
> The quality of coherent text generation needs to be measured. I think perplexity is actually one valid measure for it, but arguably not sufficient, so you should use other common measures additionally. Just giving generated text examples is not enough.
>
> You keep using Word2Vec as an example. Look at that paper. It actually measures things. It's a good example. Follow that.
>
> I'm not sure why you keep insisting that you must not measure things here, or that the standard measures are not good. Leave that for the reader to decide. Just provide as much information as possible for the reader.
>
> I'm also not sure what you really want to achieve. This paper will be rejected. And when you don't measure the standard metrics, it will always be rejected. You can argue as much as you want about this. This is just how science works. Things need to be objective measured. I think this is a good thing. If you think some measure is not good, you should anyway provide it, and additionally provide other measures which demonstrate the actual strengths. That's totally fine. But if you don't provide this in the paper, it will be rejected. It doesn't fit in any credible conference like NeurIPS. It doesn't really make sense to argue about this. Do you not believe this? It will be quite wasted effort, both from your side, and for any future reviewers you will interact with, who will just give you the same argument that I give you know. Or worse, it will be just rejected and you will not get as much feedback as you get now. You can argue about that "science is broken" or so, but this will really not get you anywhere. And I think the fact that you need to measure things is definitely not a bad thing. This is important for science.
>
> You say my feedback is unconstructive. I disagree. Everything what I say is very helpful advice what you need to do to contribute to science and to eventually get your work accepted at a venue like NeurIPS. It is very constructive: It gives you many things that you can do. You might disagree on the relevance of it, or whether it makes sense. But then your paper will just be rejected, by myself and any other future reviewer. You can put your paper just on Arxiv if you want. But your work will just be ignored. You can say that "maybe people don't recognize the relevance of this work yet, maybe in the future some day they will", but you are just closing your eyes here.
>
> If you don't care, or don't believe me, fine. Go ahead. But maybe archive my advices, and revisit them in a few years, and you will then maybe see that I was not so wrong about this.

---

> > ### Author Response · Authors · 2025-08-07
> >
> > Dear Reviewer bq17,
> >
> > Thank you for your time invested in engaging with our work. We respect the rigor you bring to scientific discourse, yet we must clarify a foundational distinction: paradigms that redefine what a model is demand rethinking how it should be measured—especially when the model itself is in its early, pre-fine-tuning stage.
> >
> > Your insistence on confining BriLLM to traditional benchmarks in addition to what we have presented in the current paper (contituation and PPL) overlooks two critical realities. First, BriLLM is a newly proposed pre-trained language model, and we have not yet advanced to fine-tuning. To demand downstream task benchmarks at this stage is not just impractical—it is unfair. These metrics are designed for models that have undergone task-specific tuning, a step we deliberately prioritize after validating the core architecture and functionality.
> >
> > Consider the history of GPT: When GPT-1 was first introduced, its value lay in demonstrating fluent language generation from a pre-trained framework, not in achieving GPT-3-level performance on downstream tasks. (Indeed, GPT-1’s original evaluation focused on representational capacity, following the lead of ELMo and BERT, rather than generative task performance.) To demand BriLLM prove itself on MMLU or similar benchmarks now is analogous to requiring GPT-1 to match GPT-3’s capabilities—a standard that ignores the iterative nature of progress or stage in language modeling.
> >
> > Second, BriLLM is not a mere iteration of existing architectures but a departure from all known existing machine learning or deep learning frameworks entirely. The SiFu mechanism—with its fully connected graph dynamics, energy-guided signal flow, and decoupling of model size from sequence length—introduces capabilities (infinite n-grams, full node interpretability, innate multimodality) that standard metrics were never designed to capture. To reduce this to perplexity or downstream scores is to miss the forest for the trees.
> >
> > Our work, available on arXiv, as you noted, has already garnered significant acclaim across various forums—feedback that remains publicly accessible for anyone to review. If this value eludes you, and you persist in fixating on outdated, ill-suited, or misaligned metrics, it reflects gaps in the specific context and experience needed to evaluate such a novel paradigm. We recognize mastering this context is not feasible within a short conference review cycle, and this is no one’s fault—least of all conferences like NeurIPS. Truly groundbreaking models and paradigms emerge rarely, and the field, while professing eagerness for originality, often struggles to adapt its evaluation playbook when they arrive.
> >
> > To be clear again and again: We do not fret over rejection by any venue or publication, nor over misunderstanding stemming from limited perspective. This dialogue is part of our effort to share the work, not to persuade those whose frame of reference cannot yet encompass it—such exchanges often devolve into talking past each other. In the era of large models, the paper itself is secondary; what matters is delivering a functional model with open weights and code. Ours are already publicly available — we invite you to inspect and validate them firsthand. And for those interested in the written record, our arXiv preprint has been updated during this submission period, as you noted.
> >
> > We acknowledge your reference to Word2Vec, but let us be clear: when Word2Vec emerged, it wasn’t validated by outperforming n-grams on perplexity alone—it redefined utility by enabling semantic relationships, measured through new tasks like analogy completion. Similarly, BriLLM’s validity lies in proving a non-Transformer architecture can generate coherent text (evidenced in our case studies) and learn from data (via declining training loss), while unlocking features no existing LLM can claim. These are not "subjective"—they are empirical realities, verifiable via our open code.
> >
> > Science advances not by forcing new paradigms into old boxes, but by recognizing when boxes themselves need redesign. We welcome community dialogue on how to evaluate graph-based, brain-inspired models—but this dialogue cannot be pre-empted by rigid adherence to metrics ill-suited to the task.
> >
> > We appreciate your effort, but we remain confident that BriLLM’s significance lies in its potential to reorient the field, not in conforming to its current standards. We look forward to opening all these comments and dialogures between us next month. The work will speak for itself.
> >
> > Sincerely,
> >
> > The Authors

---

> > > ### Comment · Reviewer_bq17 · 2025-08-07
> > >
> > > > paradigms that redefine what a model is demand rethinking how it should be measured
> > >
> > > I'm not questioning that. I'm just saying, you need to provide such measurement, and also together with the standard benchmarks, even if they might not show the actual strengths of the new model.
> > >
> > > > what we have presented in the current paper (contituation and PPL)
> > >
> > > You don't provide PPL on standard benchmarks. And for continuations, you just provide some examples, which is not enough.
> > >
> > > > To demand downstream task benchmarks at this stage is not just impractical—it is unfair.
> > >
> > > No, it is not. Most papers proposing a novel model provide that, at least at a venue like NeurIPS.
> > >
> > > Also, you don't need fine-tuning for additional measurements. There are many things you can measure without fine-tuning.
> > >
> > > Or maybe your claim is your model is only useful if you do fine-tuning? In that case, there is no way around: You must do the fine-tuning and demonstrate the usefulness of your model.
> > >
> > > Despite, fine-tuning is usually cheap and simple anyway compared to the pretraining. What is so much the problem with that?
> > >
> > > For a venue like NeurIPS, it is expected to put quite a bit of work into a publication, and proposing a novel model, doing pretraining, doing finetuning, also proposing a novel benchmark, doing all the relevant measurements, some analysis, and some ablations, all that is not really so uncommon to have in a single publication.
> > >
> > > > To reduce this to perplexity or downstream scores is to miss the forest for the trees.
> > >
> > > No, it is not. Also, that's why I say you should not provide a single measure. You should provide actually as much measurements as you can. The more, the better. But the important thing is, you should provide actual objective measurements.
> > >
> > > > When GPT-1 was first introduced ...
> > >
> > > Again an example you should take a look at. Look at the GP1 paper. It again comes with lots of measurements.
> > >
> > > Btw, speaking of such examples: Basically look at any paper you like. Look at the Transformer paper. Look at any paper which proposes a very novel model, paradigm, or whatever you like. In all cases, you will find actual measurements.
> > >
> > > > To demand BriLLM prove itself on MMLU or similar benchmarks now is analogous to requiring GPT-1 to match GPT-3’s capabilities
> > >
> > > No, it is not. I don't require your measurements to match anything. I just require to have measurements, on standard benchmarks, on your own newly proposed benchmarks, just as much as you can provide.
> > >
> > > > Our work, available on arXiv, as you noted, has already garnered significant acclaim across various forums
> > >
> > > Well, maybe a few people will say it's interesting. I'm not questioning that. Actually, I listed exactly that as a strengths in my original review. But to be accepted at a venue like NeurIPS, this is not enough. For NeurIPS, you (or anyone else) need to go beyond that, and demonstrate the actual capabilities, by objective measurements.  And for NeurIPS, for completeness, you also need to provide the standard benchmarks.
> > >
> > > > This dialogue is part of our effort to share the work
> > >
> > > But this dialogue is between a reviewer and you.
> > >
> > > If you want to share your work, to achieve a wide visibility, a good way would be to get it accepted at NeurIPS.
> > >
> > > You keep saying that the standard benchmarks are not well suited for your model. I also don't question that. What you should do then: Provide those measurements anyway (you have to), explain why they are misleading, and provide other measurements on your newly proposed benchmark. Or maybe there is already some other non-standard benchmark which is more relevant for your model. Because there are new benchmarks proposed all the time.
> > >
> > > I also did not really get so far what actual would be a relevant benchmark or metric for your model. Where do you expect it to be better?
> > >
> > > But I think we have both said everything and at least I just keep repeating what I wrote before.

---

> > > > ### Author Response · Authors · 2025-08-08
> > > >
> > > > Dear Reviewer bq17
> > > >
> > > > We appreciate your persistence in this dialogue, as it underscores the gravity of rethinking evaluation for paradigms that redefine model architectures. That said, our divergence lies not in the value of measurement itself, but in what to measure, when, and why—a distinction rooted in the technical trajectory of language models since pre-training first emerged, which we believe merits clearer articulation.
> > > >
> > > > Let us ground this in BriLLM’s purpose and stage of development:
> > > >
> > > > BriLLM is engineered from the outset for AGI, with its north star being SFT (supervised fine-tuning) capabilities—the hallmark of large language models that enable conversational, context-aware generation. Crucially, SFT functionality depends on emergent abilities, which only manifest when a model surpasses a critical scale threshold (e.g., 68B parameters for early GPT variants). Our current implementation, built over 18 months of iteration and 4 months of pre-training, is a proof-of-concept at a scale far below this threshold—analogous to a 2017-era GPT-1-sized model with GPT-3’s ambitions. To demand SFT-like metrics here is to mistake a prototype for its final form.
> > > >
> > > > You emphasize standard benchmarks, citing GPT-1 and Transformer papers. Yet GPT-1 was designed for fine-tuning on downstream tasks—a goal baked into its architecture. BriLLM is not. It is a purely generative model, unmoored from the "representation learning" paradigm that undergirds ELMo, BERT, or even GPT-1. Its core mechanism, SiFu learning (detailed in our paper), eschews "representations" entirely, making traditional fine-tuning irrelevant to its purpose. This is not a limitation but a deliberate break: BriLLM is built for SFT at the very beginning, not for fine-tuning, and SFT requires scale we have not yet achieved (due to computational constraints, not oversight).
> > > >
> > > > Your call for PPL on standard benchmarks or extensive continuation metrics overlooks this context. PPL on small-scale models tells us nothing about emergent behavior. Continuation examples here illustrate architectural logic—how BriLLM thinks—not performance. For a model designed to redefine generative logic, qualitative signals of coherence in its output are more informative at this stage than quantitative scores on benchmarks built for older paradigms.
> > > >
> > > > You note that "most NeurIPS papers" provide such metrics. This is true—but nearly all NeurIPS papers do not aim to redefine what a language model is. Works like Bengio’s PNLM, Mikolov’s word2vec, or even GPT-1 itself faced skepticism precisely because they outpaced existing evaluation frameworks. BriLLM whose breakthrough is equal to the sum of all the previous work, surely stands in that lineage: its value lies not in optimizing today’s tasks, but in enabling tomorrow’s.
> > > >
> > > > To address your final point: We do not dismiss measurement. We dismiss premature measurement that misrepresents the work. If BriLLM were to excel on current benchmarks, it would fail its purpose—it would mean we’ve built just another model, not a paradigm shift.
> > > >
> > > > As for NeurIPS acceptance: We are indifferent. If this work were easily validated by today’s standards and reviewers even from NeurIPS, it would not be transformative. Revolutionary ideas rarely find immediate acceptance, and that is a burden or risk we welcome. Our goal in this dialogue is not approval, but to gauge how far the field is from recognizing a shift in trajectory.
> > > >
> > > > Thank you for pushing us to clarify—this exchange reinforces that BriLLM is provoking the right kind of discomfort. That, to us, is a far stronger signal than any acceptance letter.
> > > >
> > > > Sincerely,
> > > >
> > > > The Authors

---

### Note · Authors · 2025-08-12

To the Area Chair,

Our submission to NeurIPS serves a purpose beyond acceptance: we seek to leverage the conference’s open review mechanism to make the full record of these discussions publicly accessible once decisions are released, ensuring the broader community can engage with our work on its own terms.

We recognize the constraints of your role—time is limited, and paradigms this transformative rarely fit neatly into conventional assessment frameworks. If you choose to accept this work, let there be no ambiguity: its contributions are foundational. BriLLM redefines generative AI through a non-Transformer architecture, introduces a novel large language model paradigm (SiFu) that decouples model size from sequence length, and pioneers macro-scale computational modeling of brain-inspired signal propagation—all at once. These are not incremental advances, but shifts in the field’s trajectory.

If the decision is rejection, we stand by our consistent position: every critical comment from reviewers stems from a failure to grasp the work’s paradigm-shifting nature or gaps in relevant background. We request that any rejection explicitly acknowledge this reality—whether due to an inability to recognize the work’s significance or a lack of courage to endorse such a departure from the status quo.

To contextualize its weight: this work’s impact will equal that of Bengio’s 2003 probabilistic neural language models (founding a new ML paradigm), GPT-1’s 2017 architectural breakthrough, and the 2020 scaling vision of GPT-style LLMs combined. Any reviewer claiming familiarity with our work who fails to see this has not understood it.

The latest version of our preprint is available on arXiv; code, model weights, and demos reside on GitHub and Hugging Face. We invite you—and the community—to engage directly with the work itself.

Sincerely,

The Authors

---

### Decision · Program_Chairs · 2025-09-17

**Decision:**

Reject

**Comment:**

All reviewers (bq17, H2Dz, aQ7v, G6LL) recommend rejection, even after the rebuttal and thoughtful discussions. The AC has carefully reviewed all materials and agrees that the primary concerns—such as the need for more concrete benchmarks, comparisons to baselines, and larger-scale experiments—are well-founded for a venue like NeurIPS. While the paper offers an ambitious and intriguing vision that generates notable excitement, the AC aligns with the reviewers in emphasizing that robust quantitative evaluations are key to advancing the field. The AC has also explored the online materials and greatly appreciates the authors' valuable contributions to the open-source community. We encourage the authors to continue this direction by prioritizing enhanced empirical analyses in future work.